# Predicting mechanisms of action at genetic loci associated with discordant effects on type 2 diabetes and abdominal fat accumulation

Yonathan Tamrat Aberra[1,2]*, Lijiang Ma[3], Johan LM Björkegren[3,4], Mete Civelek[1,2]*

[1]Department of Biomedical Engineering, University of Virginia, Charlottesville, United States; [2]Center for Public Health Genomics, University of Virginia, Charlottesville, United States; [3]Department of Genetics and Genomic Sciences, Icahn School of Medicine at Mount Sinai, New York, United States; [4]Department of Medicine, Karolinska Institutet, Huddinge, Stockholm, Sweden

*For correspondence:
ya8eb@virginia.edu (YTamratA);
mete@virginia.edu (MC)

Competing interest: The authors declare that no competing interests exist.

**Abstract** Obesity is a major risk factor for cardiovascular disease, stroke, and type 2 diabetes (T2D). Excessive accumulation of fat in the abdomen further increases T2D risk. Abdominal obesity is measured by calculating the ratio of waist-to-hip circumference adjusted for the body-mass index (WHRadjBMI), a trait with a significant genetic inheritance. Genetic loci associated with WHRadjBMI identified in genome-wide association studies are predicted to act through adipose tissues, but many of the exact molecular mechanisms underlying fat distribution and its consequences for T2D risk are poorly understood. Further, mechanisms that uncouple the genetic inheritance of abdominal obesity from T2D risk have not yet been described. Here we utilize multi-omic data to predict mechanisms of action at loci associated with discordant effects on abdominal obesity and T2D risk. We find six genetic signals in five loci associated with protection from T2D but also with increased abdominal obesity. We predict the tissues of action at these discordant loci and the likely effector Genes (eGenes) at three discordant loci, from which we predict significant involvement of adipose biology. We then evaluate the relationship between adipose gene expression of eGenes with adipogenesis, obesity, and diabetic physiological phenotypes. By integrating these analyses with prior literature, we propose models that resolve the discordant associations at two of the five loci. While experimental validation is required to validate predictions, these hypotheses provide potential mechanisms underlying T2D risk stratification within abdominal obesity.

## Editor's evaluation

This study presents a valuable finding on five genome-wide association study loci displaying discordant effects on T2D and abdominal obesity. The evidence supporting the claims of the authors is solid, although further experiments are required to describe the mechanisms through which a genetic variant can confer increased abdominal obesity while offering protection from T2D risk. The work will be of interest to researchers working within the fields of variant-to-function analysis and endocrinology.

## Introduction

Metabolic syndrome (MetSyn) is a cluster of dysregulated metabolic conditions that tend to occur together to increase the risk for cardiometabolic disorders such as type 2 diabetes (T2D) (*Lusis, 2006*).

This cluster includes insulin resistance (IR), abdominal obesity, elevated serum triglycerides (TG) levels, low high-density lipoprotein cholesterol levels, as well as elevated systolic and diastolic blood pressure. Obesity, or the excessive accumulation of fat that presents a risk to health, is a major contributor to MetSyn (*Lusis, 2006*; *Feero et al., 2010*). Obesity, which is typically defined as a body mass index (BMI) above 30, has reached unprecedented levels of prevalence, and its role as a central regulator of disease risk makes it an appealing therapeutic target (*Feero et al., 2010*). Several recently developed T2D therapeutics have even successfully targeted obesity; SGLT2 inhibitors and GLP-1 agonists have been reported to result in a 2–6 kg reduction of body weight and reduced IR (*Brown et al., 2021*).

Despite the promise of these obesity-centered therapeutic strategies, there has also been a growing body of evidence describing a rare phenotype known as metabolically healthy obesity (MHO) (*Blüher, 2020*). MHO describes a group of phenotypes in which individuals with obesity are protected from adverse metabolic effects (*Smith et al., 2019*). While no formal definition of MHO exists, it is often described as either obesity with less than three components of MetSyn, or obesity without IR as computed by the homeostasis model assessment of insulin resistance (HOMA-IR) (*Smith et al., 2019*). Mechanisms proposed to mediate this include depressed ectopic fat accumulation, subcutaneous adipose tissue (SAT) expansion plasticity, and shifts in fat storage from the abdomen to the legs (*Blüher, 2020*; *Smith et al., 2019*; *Loos and Kilpeläinen, 2018*). In recent years the ability of abdominal obesity to mediate cardiometabolic disease risk has gained attention. People with MHO have less intra-abdominal fat accumulation compared to people with metabolically unhealthy obesity (*Klöting et al., 2010*; *Karelis et al., 2005*; *Chen et al., 2015*; *Jennings et al., 2008*; *Hayes et al., 2010*; *Koster et al., 2010*). Intra-abdominal fat accumulation can be approximated through the ratio of waist-to-hip circumference (WHR) adjusted for BMI (WHRadjBMI). WHRadjBMI is a causal factor that increases susceptibility for T2D, but the genetic and molecular mechanisms underlying fat distribution remain largely unknown (*Emdin et al., 2017*; *Gill et al., 2021*; *Li et al., 2021*). Understanding the mechanisms mediating WHRadjBMI, MHO, and T2D is critical to our understanding of disease pathogenesis and to clinical strategies to treat MetSyn.

Most of the genetic mechanisms of MHO described have been associated with increased BMI without increased disease risk. For example, the missense variant rs373863828 in CREB3 regulatory factor has been shown to increase BMI without a corresponding increase in HOMA-IR and circulating TG, or a decrease in circulating adiponectin (*Minster et al., 2016*). Ob/ob mice with overexpression of adiponectin but lacking in leptin are shown to accumulate considerable fat mass without a corresponding increase in insulin sensitivity (*Kim et al., 2007*). In contrast, the genetic loci associated with increased WHRadjBMI but without increased disease risk have not yet been described. To date, all genes that have been shown to increase fat accumulation into abdominal fat depots have also been shown to increase the risk for T2D (*Fathzadeh et al., 2020*; *Small et al., 2018*; *Yang et al., 2022*; *Gesta et al., 2011*; *Loh et al., 2020*; *Loh et al., 2015*).

As complex traits with both environmental and genetic risk factors, abdominal obesity and T2D have been the subject of multiple genome-wide association studies (GWAS). While GWAS have identified hundreds of genetic loci associated with abdominal obesity and T2D, moving from association to mechanism at a locus is not trivial. The use of colocalization analysis (COLOC), which identifies loci that contain shared genetic architecture for multiple traits of interest, can inform mechanistic hypotheses moving from association to function by integrating data from multiple studies (*Hukku et al., 2021*; *Wallace, 2013*; *Wallace et al., 2012*; *Wallace, 2020*). For example, the colocalization of a GWAS signal with genetic regulation of genes at quantitative trait loci (QTLs) implies a mechanistic relationship between the regulated gene and GWAS trait (*Hormozdiari et al., 2016*). Another recently developed approach named Tissue of ACTion scores for Investigating Complex trait-Associated Loci (TACTICAL) (*Torres et al., 2020*) incorporates gene expression data, and epigenetic annotations with GWAS associations to predict the causal eGenes and tissues of action at GWAS loci. These methods have been used to inform data-driven mechanistic predictions at GWAS loci that have been experimentally validated and can recall previously validated loci as positive controls.

To advance the understanding of mechanisms linking body fat distribution to T2D risk, independent of overall obesity, we used COLOC and TACTICAL to predict the mechanisms of action at genetic loci associated with both T2D and WHR, both adjusted for the BMI. Using the most recent GWAS summary statistics, QTL summary statistics, tissue-specific gene expression data, and high-resolution epigenetic annotations, we predicted the shared genetic architecture of T2DadjBMI and WHRadjBMI

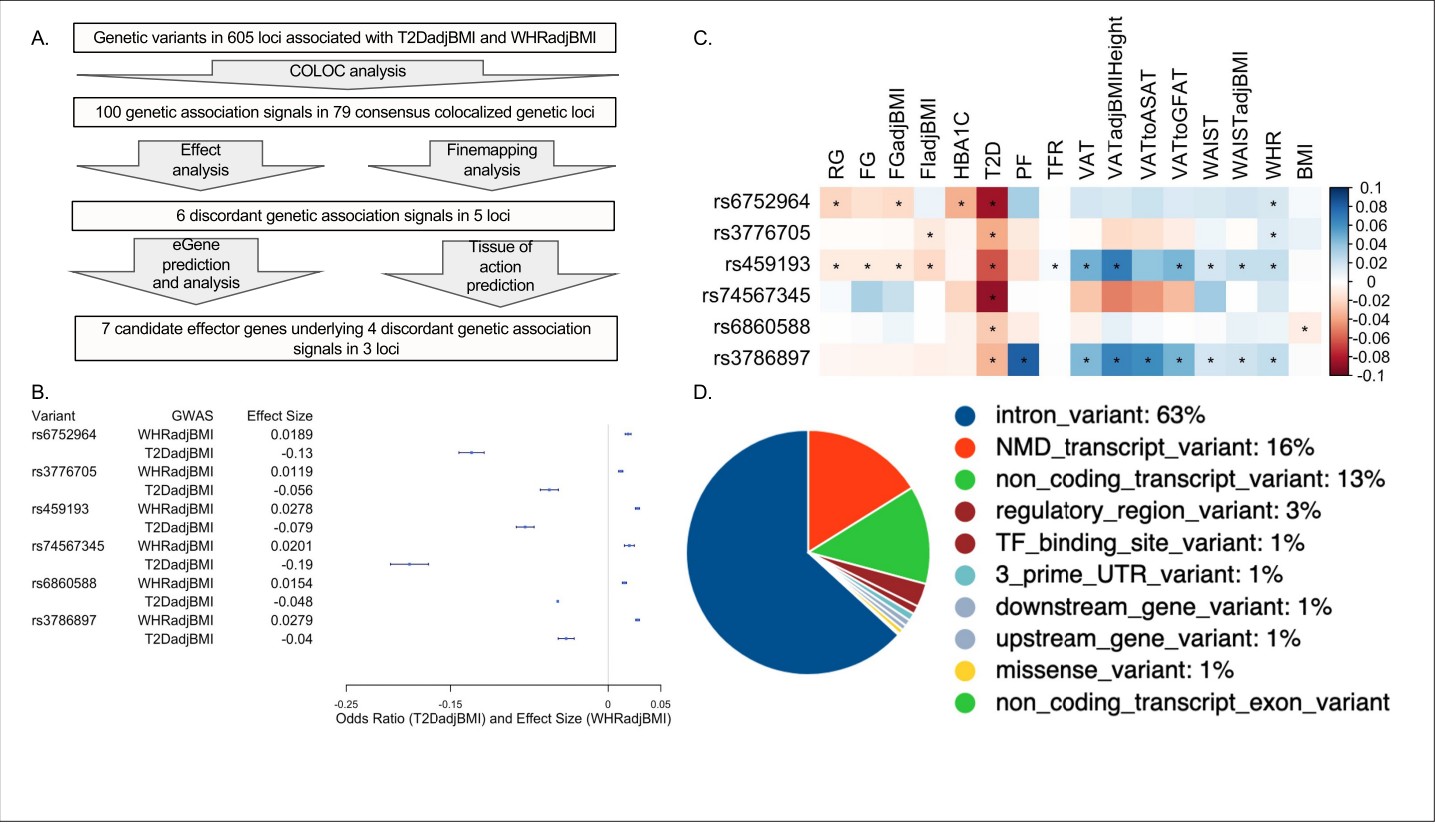

**Figure 1.** Analysis summary and discordant variant characteristics. (**A**) Summary of analysis pipeline and generated results. Details of data sources are available in *Supplementary file 1*. (**B**) Effect size (WHRadjBMI) and odds ratio (T2DadjBMI) of lead genetic variant at discordant association signals. (**C**) Phenome-wide association study (PheWAS) of lead discordant genetic variant effect sizes on glycemic and anthropometric traits. From left to right: random glucose (RG), fasting glucose (FG), FG adjusted for body mass index (BMI) (FGadjBMI), fasting insulin adjusted for BMI (FIadjBMI), glycated hemoglobin (HBA1C), pancreatic fat percentage (PF), trunk fat ratio (TFR), visceral adipose tissue (VAT), VAT adjusted for BMI and height (VATadjBMIHeight), VAT to abdominal subcutaneous adipose tissue (VATtoASAT), VAT to gluteofemoral fat (VATtoFGAT), waist circumference, waist circumference adjusted for BMI (WCadjBMI), waist-to-hip ratio (WHR), and BMI. (**D**) Variant effect prediction of 99% credible set variants in discordant genetic loci.

The online version of this article includes the following source data for figure 1:

**Source data 1.** Genetic, transcriptomic, and epigenomic data sources in *Figure 1*.

at 79 genetic loci. Here, we present the identification of five loci that contained association signals with discordant effects on abdominal fat and T2D risk, meaning that the allele of the lead variant associated with protection from T2D was associated with increased abdominal fat accumulation. We predicted the eGenes and tissues of action at these five loci and explored the relationship between adipose eGene expression with cellular and physiological phenotypes. Here, we provide data-driven hypotheses about predicted candidate causal eGenes at GWAS loci with associations that recall metabolically healthy abdominal obesity.

## Results

### Colocalization analysis of genetic loci associated with T2D and body fat distribution predicts colocalization of discordant T2DadjBMI and WHRadjBMI association signals at six association signals

To identify genetic loci which contained pleiotropic association signals for both T2DadjBMI and WHRadjBMI, we performed colocalization analysis (*Figure 1A*). This analysis yielded 79 genetic loci where a single variant was significantly associated with both T2DadjBMI and WHRadjBMI. We obtained the 99% credible set of variants in colocalized loci (*Supplementary file 1*) and discovered

the presence of 143 variants in five loci associated with discordant effects on T2DadjBMI and WHRadjBMI. We also discovered 851 SNPs in 73 loci with the expected concordant effects on both traits. Although almost all of the representative lead discordant variants reached genome-wide significance, two associations reached nominal significance (p<5e-05). Because of recent work demonstrating that even variants with only nominal and local significance in GWAS can also have functional relevance to GWAS traits, we included variants prioritized in the 99% credible set but with only nominal significance (*Li et al., 2020a*). We then performed fine-mapping of the causal variants in each locus containing a discordant association signal while relaxing the assumption of a single causal variant per locus. In four of the five loci, this fine-mapping recalled only one likely candidate causal signal. In the 5q21.1 locus, SuSiE identified a secondary association signal that was also associated with discordant effects on T2DadjBMI and WHRadjBMI (*Figure 1B* and *Supplementary file 2*). To parse the associations between specific components of WHRadjBMI, including WC, HC, WHR, and BMI, with both T2D and T2DadjBMI, we performed multi-trait colocalization analysis with Hyprcoloc of the associations at discordant loci (*Supplementary file 3*). At three of the five discordant loci, the discordant association signals were also colocalized with WHRadjBMI component traits waist circumference (WC) and WHR.

We next investigated the genetic and physiological consequences of discordant variants. We performed a phenome-wide association study (PheWAS) for anthropometric and glycemic traits with the most highly powered GWAS available (*Figure 1—source data 1*; *Costanzo et al., 2023*). We used the most highly powered GWAS or GWAS meta-analysis for each trait included in our PheWAS and queried the summary statistics for the associations of each lead discordant variant (*Figure 1C*). This query revealed consistent significant associations with discordance across anthropometric and glycemic traits in each locus. At the association signal in the 5q11.2 region, association signals exemplified this metabolic discordance. Represented by genetic variant rs459193, the association signal was associated with increased abdominal obesity in nearly every metric, but also with protection from

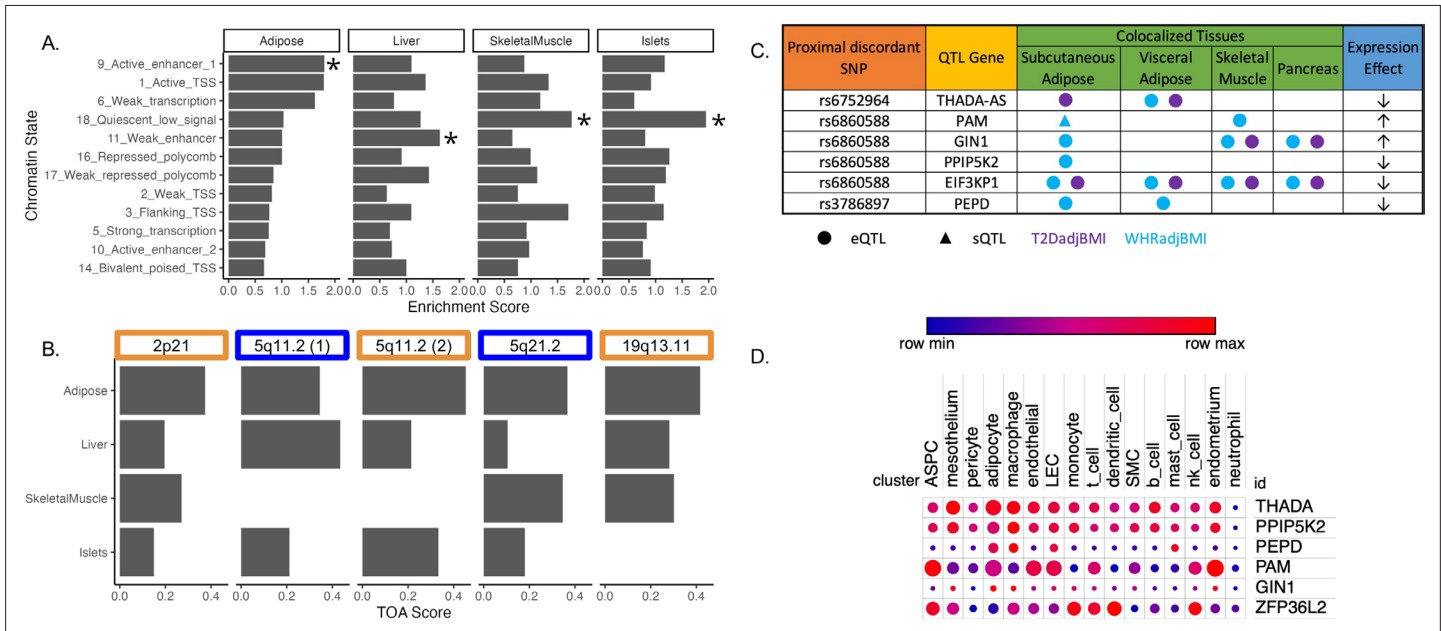

**Figure 2.** Predicting functional tissues and effector genes at discordant loci. (**A**) Tissue-specific enrichment of chromatin states of variants in the 99% credible set of colocalized variants. (**B**) Tissue of action scores for association signals in the five discordant loci. Orange coloration indicates predicted adipose tissue of action at the locus, and blue coloration indicates shared tissue of action assignment at the locus. (**C**) Summary table of the expression quantitative trait loci (eQTL) and splicing QTL (sQTL) colocalizations with waist-to-hip circumference adjusted for the body mass index (WHRadjBMI) and T2DadjBMI for discordant loci. The expression effect direction is with respect to the protective type 2 diabetes allele. (**D**) Expression of predicted effector genes in discordant loci across cell types. From left to right: adipocyte progenitor stem cells (ASPC), lymphatic endothelial cells (LEC), smooth muscle cells (SMC), and natural killer cells (nk). Data was obtained from *Emont et al., 2022*.

The online version of this article includes the following source data for figure 2:

**Source data 1.** Genetic, transcriptomic, and epigenomic data sources used in *Figure 2*.

T2D in nearly every metric. At all lead discordant variants, effects were consistent with a phenotype of increased abdominal obesity but protection from T2D.

We then queried the variant effect predictor (VEP) to discover genetic variant annotations (*McLaren et al., 2016*; *Figure 1D* and *Supplementary file 4*). VEP predicted that discordant variants overwhelmingly lie in noncoding regions of the genome, with only one missense variant in a coding region. Because the vast majority of discordant variants lie in noncoding regions, it is likely their function lies in altering genetic regulation of proximal genes (*Civelek and Lusis, 2014*). Therefore, we investigated the coincidence of these discordant variants with the genetic regulation of proximal genes with functional prediction methods.

## Integration of molecular QTLs and genomic annotations to predict functional genes in tissues of action at discordant genetic loci

To investigate the role of eGenes in physiological phenotypes and cellular phenotypes, we evaluated the correlation of adipose tissue eGene expression and T2D-relevant phenotypes since these correlations can reveal biologically relevant functional relationships (*Civelek et al., 2017*). To predict the genes and tissues of function at discordant loci, we used publicly available multi-omic data from metabolically relevant tissue-specific resources to predict functional mechanisms underlying associations. We first interrogated where the 143 discordant variants in the credible set were located in relation to tissue-specific chromatin state data in pancreatic islet, adipose, liver, and skeletal muscle tissues (*Kim et al., 2007*). We computed the enrichment of colocalized association signals in various chromatin state annotations in each of these tissues (*Figure 2A*). We noted the specific enrichment of adipose tissue chromatin states of high activity, such as active transcription start sites, enhancer regions, and areas of transcriptional activity. For every other tissue, the leading annotations represented areas of decreased transcriptional activity. We additionally queried 3D chromatin data for discordant variant enhancer/promoter contact but did not find any significant interactions (*Figure 2—source data 1*). We then used these enrichment scores, chromatin states, and gene expression data to predict the functional tissues at each colocalized locus (*Supplementary file 5*). We predicted that adipose tissue was classified as the candidate tissue of action (TOA) at three loci, and skeletal muscle and liver tissue shared classification with adipose tissue at the remaining two discordant loci (*Figure 2B*).

To predict effector genes (eGenes) regulated by discordant variants, we next predicted the colocalization of QTLs with the WHRadjBMI and T2DadjBMI GWAS. Colocalization of a GWAS association signal with a genetic regulatory association signal can be used to prioritize mechanisms underlying association. We obtained expression QTL (eQTL) and splicing QTL (sQTL) summary statistics from multiple cohorts and tissue groups (*Figure 2—source data 1*). We extracted eQTL summary statistics for all genes within 1 Mb of the lead variant of all discordant colocalized loci from adipose, pancreatic, skeletal muscle, and liver tissues. We extracted sQTL summary statistics for all genes within 1 Mb of the lead variant of all discordant colocalized loci for adipose tissue data that was available. We used Summary-based Mendelian Randomization (SMR) and Coloc.abf to perform GWAS-QTL colocalization and used the framework developed by Hukku et al. to reconcile the results of SMR and Coloc.abf. In this framework, colocalization found using Coloc.abf but not with SMR potentially represents signals with horizontal pleiotropy, whereas colocalization found through SMR but not through Coloc.abf potentially represents locus-level colocalization (*Hukku et al., 2021*). Colocalization found using both methods represents the identification of candidate causal effector transcripts. Our colocalization analysis revealed seven candidate causal effector transcripts at three of the five discordant loci (*Figure 2C*). With Coloc.abf, we predicted four putative eGenes in these two loci. At the 2p21 locus, we predicted *THADA-AS* (SAT, VAT) to be the sole eGene. At the 5q21.1 locus, we predicted *GIN1* (SAT), *PAM* (SAT & SKM), and *PPIP5K2* (SAT) to be the eGenes. The association signal at rs6860588 was associated with a novel alternative splicing isoform of *PAM* in SAT, which skips the 14th exon. Using SMR, we predicted four eGenes at two discordant loci. At the 5q21.1 locus, we predicted that the genetic association signal represented by rs6860588 was also associated with the regulation of *EIF3KP1* (SAT, VAT, SKM, PANC), *PPIP5K2* (SAT), and *GIN1* (SAT, SKM, PANC). At the discordant association signal in the 19q13.11 locus, we predicted that the genetic association signal represented by variant rs3786897 was also associated with the regulation of *PEPD* (SAT, VAT). As the colocalization transcripts *GIN1* and *PPIP5K2* were replicated with both methods (*Supplementary files 6 and 7*), these represent high-confidence predictions of potentially causal effector transcripts underlying the

genetic association with discordance in the 5q21.1 locus. We queried white adipose tissue single-cell RNA sequencing data (*Emont et al., 2022*) for discordant association signal eGenes and found that eGenes were expressed in adipocytes and adipocyte progenitor stem cells (ASPCs) (*Figure 2D*). Because body fat distribution associations are driven by ASPCs and adipocytes in adipose tissues (*Lu et al., 2016*; *Locke et al., 2015*; *Hansen et al., 2023*), we reasoned that exploring adipose expression data could help to explain discordant associations. This multi-omic data enabled us to make high-confidence consensus predictions of tissues and eGenes of action at discordant loci.

## Adipose gene expression analysis of discordant loci eGenes reveals dynamic expression in adipogenesis and relationships with metabolic physiology

To investigate the role of eGenes in physiological phenotypes and cellular phenotypes, we then evaluated the gene expression dynamics of eGenes in adipose tissue. Correlations between relevant tissue gene expression and metabolic phenotypes can reveal biologically relevant functional relationships (*Civelek et al., 2017*). We used SAT transcriptomic data from the 426 men of the METSIM cohort to investigate how adipose tissue expression of discordant locus eGenes was related to 23 metabolic phenotypes underlying T2D and abdominal fat accumulation (*Figure 3—source data 1*; *Brotman et al., 2022*; *Laakso et al., 2017*). We extracted adipose tissue gene expression data for eGenes. Gene expression data were available for six of the seven eGenes. We additionally extracted splice junction expression data for the only gene with a colocalized splice junction, *PAM*. We then computed the biweight midcorrelation of transcript counts or splice junction counts with 23 metabolic phenotypes. We found significant (false discovery rate [FDR] <0.05) correlations of adipose tissue gene expression of three genes with 13 phenotypes (*Figure 3A*). We found that adipose tissue expression of *THADA-AS*, *PEPD*, and *GIN1* was significantly correlated with inflammatory, glycemic, and anthropometric phenotypes. SAT *THADA-AS* expression was positively correlated with IR, abdominal fat accumulation, and serum triglyceride levels, but with higher levels of plasma interleukin-1 receptor antagonist (IL-1RA) and C-reactive protein (CRP). IL-1RA plays a protective role in resolving inflammation (*Volarevic et al., 2010*), and elevated levels have been linked to prediabetes (*Luotola, 2022*; *Grossmann et al., 2015*). CRP has been used as a biomarker of increased inflammation in chronic diseases (*Herwald and Egesten, 2021*). The eQTL and GWAS data are associated with decreased expression of THADA-AS, which is consistent with the protection from IR in the correlation data but not with the increased abdominal obesity and inflammation. We are unable to resolve this correlation evidence with the discordance, but because the METSIM cohort was collected using single-end RNA sequencing, parsing the correlations of THADA and THADA-AS is difficult (*Li et al., 2013*). SAT expression of *GIN1* was correlated with higher plasma adiponectin. Adiponectin, secreted by adipocytes, increases insulin sensitivity, and this provides a mechanism for protection from T2D (*Achari and Jain, 2017*). This expression is consistent with the QTL and GWAS data, providing a direct potential mechanism linking the eQTL to protection from T2D. SAT *PEPD* expression was also positively correlated with plasma IL-1RA levels. The QTL at this locus is associated with decreased expression of PEPD, providing another direct potential mechanism linking the eQTL to protection from T2D. Through this correlation analysis, we were able to predict the physiological consequences of eGenes at three discordant loci.

We next evaluated if eGenes identified in adipose tissues were dynamically expressed in adipogenesis. Dynamic gene expression in adipogenesis could point to the regulatory and structural roles of eGenes in adipogenesis (*Nassiri et al., 2016*; *Anderson et al., 2020*). We obtained time series ASPC adipogenesis time course data and evaluated eGenes for dynamic expression. Gene expression data were available for five of the seven eGenes. Because the expression data was single-stranded and unable to resolve forward or reverse-strand sequences, we included the probe for *THADA* to represent *THADA-AS*. We found that all eGenes except *PPIP5K2* were dynamically expressed over a 16-day adipogenesis time course (*Figure 3B*), implying potential functional roles for these genes in regulating preadipocyte fate.

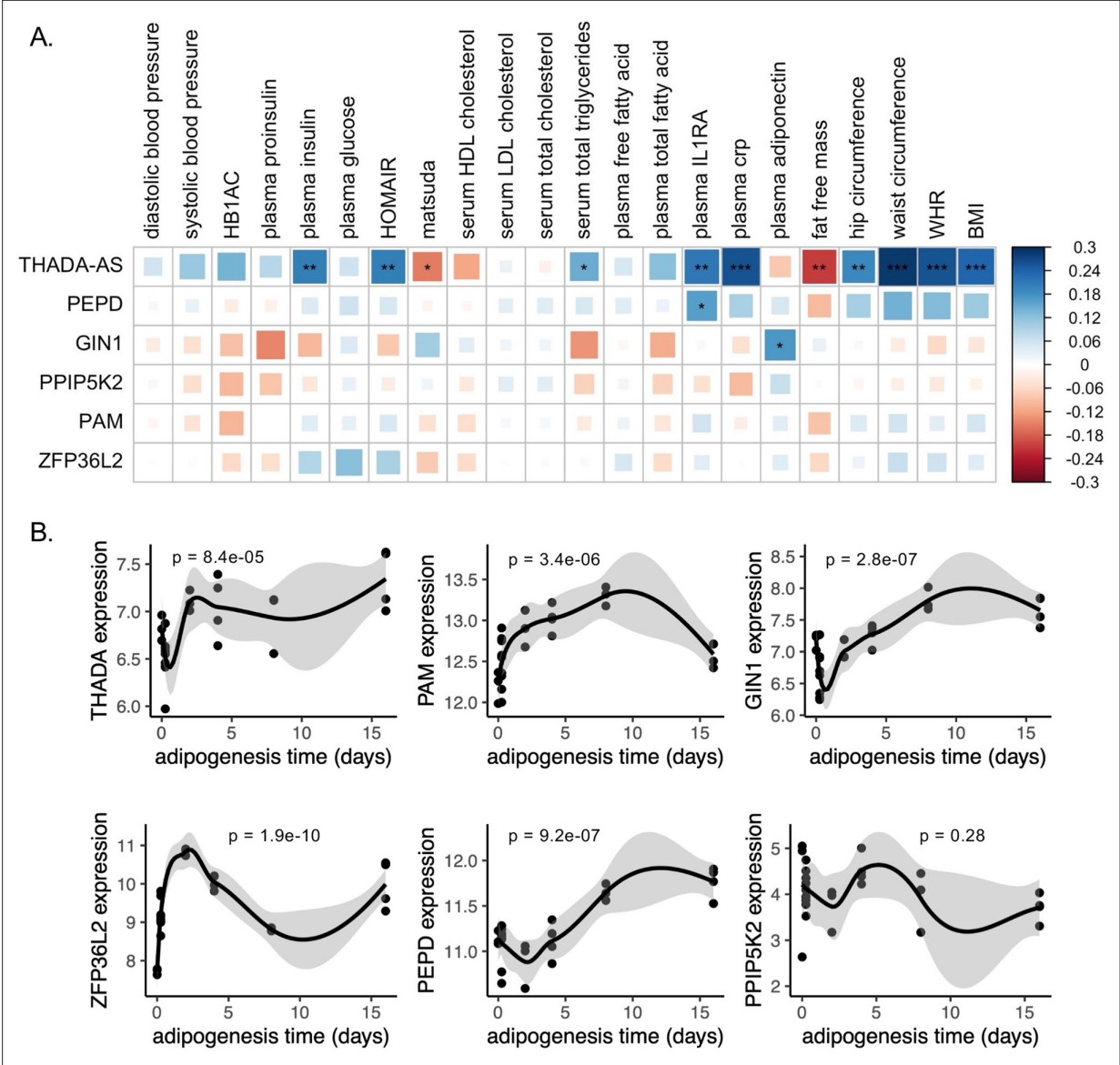

**Figure 3.** Predicted physiological and cellular effects of effector genes (eGenes) on metabolic phenotypes and adipogenesis. (**A**) Biweight midcorrelation of adipose tissue eGenes expression with metabolic phenotypes (false discovery rate [FDR] <5%). From left to right: Homeostatic model assessment of insulin resistance (HOMA-IR), high-density lipoproteins (HDL), low-density lipoproteins (LDL), interleukin-1 receptor agonist (IL1RA), C-reactive protein (CRP). (**B**) Dynamic expression of adipose tissue eGenes over 16-day adipogenesis time course in Simpson-Golabi-Behmel syndrome (SGBS) cells. We performed the likelihood ratio test (LRT) to evaluate if each gene was dynamically expressed over the time course. The p-value of the LRT is included.

The online version of this article includes the following source data for figure 3:

**Source data 1.** Genetic, transcriptomic, and epigenomic data sources in *Figure 3*.

## Integration of analysis to predict the functional genes and tissues of action at the discordant 5q21.1 locus

By predicting the mechanisms of action at discordant loci, we were able to generate specific hypotheses about the genes at each locus that underlie GWAS associations. We predicted that the causal discordant signal at the 5q21.1 locus was represented by variant rs6860588. The T allele of rs6860588 is associated with protection from T2D, increased abdominal obesity, decreased SAT expression of *GIN1*, increased SKM expression of *PAM*, decreased SAT expression of *PPIP5K2*, increased SAT expression of a *PAM* splice variant with a skipped exon 14, and decreased SAT expression of the

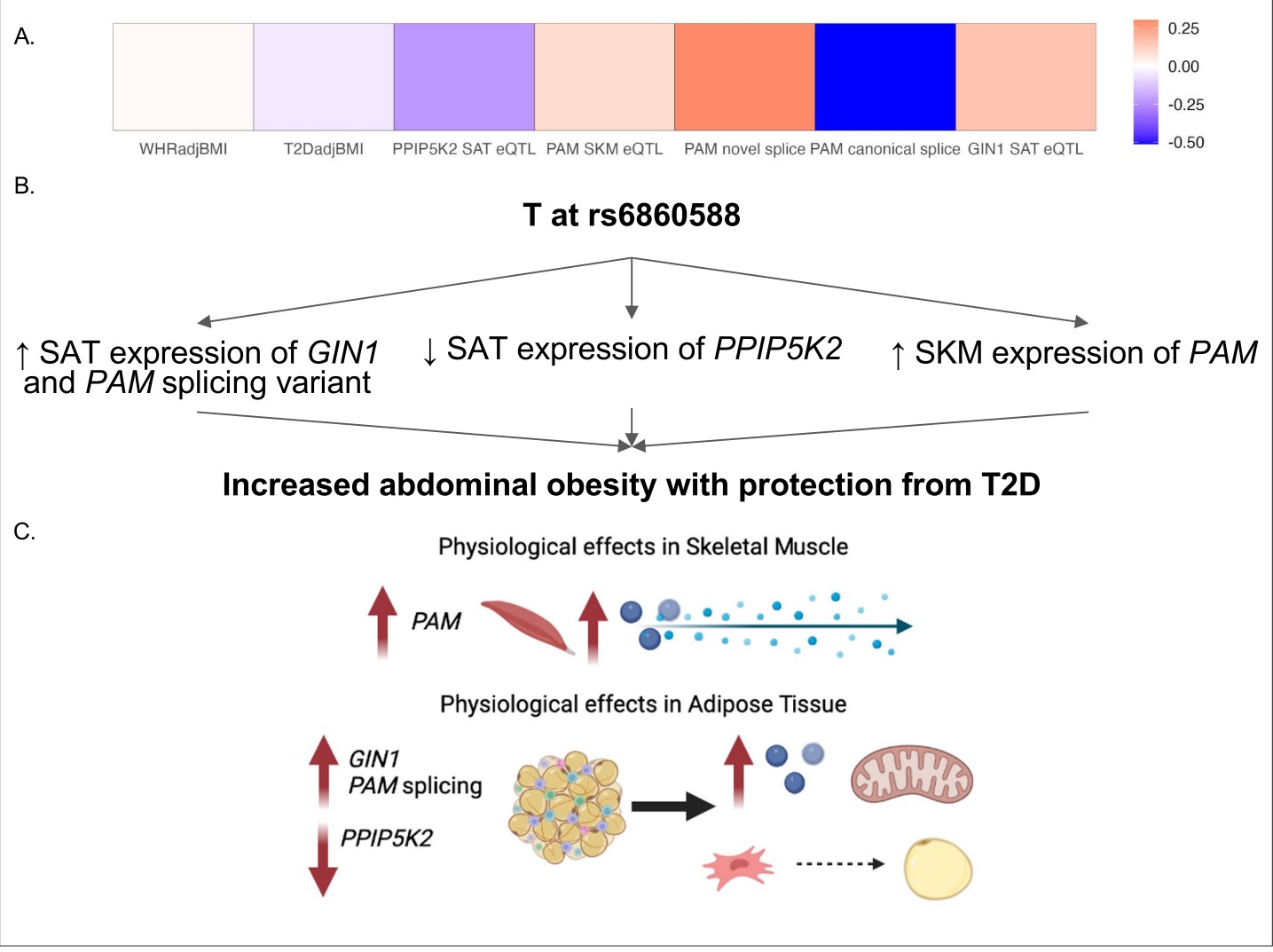

**Figure 4.** Predicted model of effects associated with T allele at rs6860588. (**A**) β of the T allele of discordant variant rs6860588 with respect to waist-to-hip circumference adjusted for the body mass index (WHRadjBMI), T2DadjBMI, and colocalized effector genes (eGenes). (**B**) Summary of associations with T allele at rs6860588. (**C**) Integrated model reconciling metabolic discordance with eGene-associated phenotypes in two tissues of action. Created with BioRender.com.

The online version of this article includes the following source data and figure supplement(s) for figure 4:

**Source data 1.** Genetic, transcriptomic, and epigenomic data sources in *Figure 4* and *Figure 4—figure supplements 1 and 2* .

**Figure supplement 1.** Discordant variant rs6860588 is associated with pleiotropic effects on gene regulation in multiple tissues.

**Figure supplement 2.** Discordant variant rs6752964 is associated with pleiotropic effects on gene regulation in multiple tissues.

canonical *PAM* splice junctions, exon 13:14 and exon 14:15 (*Figure 4A*; *Figure 4—figure supplement 1* and *Figure 4—source data 1*). While the eGenes, *PAM*, *GIN1*, and *PPIP5K2*, have not been studied in the context of obesity and metabolism, they have been studied for their function in other cell types. We found that GIN1 and PAM were dynamically expressed over the course of adipogenesis (*Figure 3B*). *GIN1* has been hypothesized to be a key regulator of energy metabolism in atria (*Li et al., 2020b*), but little is known about gypsy integrases and their molecular function. PAM facilitates C-terminus glycine residue amidation, which can catalyze protein potency (*Thomsen et al., 2018*; *Merkler, 1994*). PAM additionally has been linked to metabolic phenotypes in multiple model organisms, where its deficiency is associated with decreased peptide secretion and potency critical to insulin release, but not with increased diabetes (*Chen et al., 2020a*; *Zieliński et al., 2016*). PAM loss of function likely results in deficient peptide synthesis and secretion in adipocytes as well, and its increase

of function likely results in increased myokine signaling from skeletal muscle. Knockdown of PPIP5Ks results in decreased proliferation, increased mitochondrial mass, decreased inositol metabolism, and accelerated glycolysis in tumor cell lines (*Gu et al., 2021*; *Gu et al., 2017*; *Badodi et al., 2021*). We did not observe significant interactions between adipose *PPIP5K2* expression and adipogenesis or metabolic phenotypes, but this does not rule out a role for PPIP5K2 in the metabolic discordance at 5q21.1. Thus, we propose that the T allele at rs6860588 regulates a group of genes that promotes adipogenesis, glycolysis, and inflammation in white adipose tissue while simultaneously decreasing preadipocyte expansion and increasing skeletal muscle peptide secretion and potency (*Figure 4B*). This model is consistent with the TOA score and QTL analysis, which both predict skeletal muscle and adipose tissue contribution to the associations at the locus and reconcile the associations with abdominal obesity but protection from T2D associated with the T allele of rs6860588 (*Figure 4C*).

## Discussion

We report here the integration of multi-omic data spanning the genome, transcriptome, and epigenome to predict functional genes and tissues underlying genetic signals associated with abdominal obesity but protection from T2D. We predicted the colocalization of T2DadjBMI and WHRadjBMI association signals at 79 genetic loci. The protective allele of six association signals was associated with lower T2D risk but higher abdominal fat accumulation, independent of overall obesity (*Figure 1*). By analyzing colocalization with molecular QTLs, computing the enrichment of variants in epigenomic and genomic annotations, and comparing tissue-specific gene expression, we predicted the eGenes and tissues of action at discordant association signals (*Figure 2*). We found significant evidence that adipose tissue biology is a significant contributor at colocalized loci. We then explored the effects of eGenes expression in adipose tissue and preadipocytes on adipogenesis metabolic phenotypes (*Figure 3*) and proposed a model by which the genetic variant rs6860588 might confer protection from T2D but increased abdominal obesity (*Figure 4*).

The six genetic association signals associated with discordant metabolic phenotypes offer potential insight into the genetic mechanisms underlying risk stratification of T2D risk within abdominal obesity. While mechanisms promoting MHO have been described, most have focused on body fat distribution. Defining more mechanisms that promote MHO is critical as rates of obesity rise globally. Complicating the study of MHO is the lack of precision in its definition. Some definitions include obesity with less than three components of MetSyn, obesity with healthy HOMA-IR, or even obesity with the lack of a metabolic and cardiovascular disorder (*Blüher, 2020*). MHO has been controversial and termed an intermediate state (*Caleyachetty et al., 2017*; *Rey-López et al., 2015*; *Blüher, 2017*), but a growing body of evidence has accumulated providing evidence that genetic mechanisms influence predisposition to it. In Samoans, the common *CREBRF* coding variant rs12513649 increases BMI and overall adiposity but protects from IR (*Li et al., 2020a*). Additionally, *IRS1*, *COBLL1*, *PLA2G6*, and *TOMM40* have been associated with higher BMI but with protective lipidemic and glycemic traits (*Loos and Kilpeläinen, 2018*). The physiological functions of these genes have been proposed to involve adipose tissue caloric load capacity and body fat distribution (*Loos and Kilpeläinen, 2018*; *Lu et al., 2016*; *Kilpeläinen et al., 2011*; *Lotta et al., 2017*).

While abdominal fat accumulation is known to be one of the strongest predictors of obesity-related complications (*Emdin et al., 2017*; *Censin et al., 2019*; *Dale et al., 2017*), our findings point to mechanisms that contradict this trend. Each locus must be functionally annotated before translating the association results to the clinic. If these discordant variants are functionally annotated and fully characterized, they might have clinical utility to T2D risk allele carriers and inform personalized therapeutic strategies. Discovering mechanisms uncoupling abdominal obesity from T2D can aid in personalized therapeutic strategies and in understanding personalized risk stratification. Risk-stratified personalized obesity treatment could prioritize patients that would or would not benefit significantly from weight-loss interventions, and use genotype as a biomarker for patients who would benefit from other therapeutic strategies (*Klonoff, 2008*; *Williams et al., 2022*). Thus, the importance of personalized risk stratification for T2D will only increase as abdominal obesity becomes more prevalent. Personalized risk stratification with an understanding of specific molecular, cellular, and physiological mechanisms will aid in the prioritization of effective therapies. This investigation provided specific hypotheses linking functional genes at discordant loci to tissues of action for experimental follow-up

in vitro and in vivo. Functional characterization of the effect of these genes on insulin uptake, preadipocyte proliferation, and adipogenesis, as well as secretome characterization, will elucidate precise mechanisms through which these eGenes might contribute to the discordant association signals.

We predicted tissues and mechanisms of action at five loci containing six discordant association signals with increased abdominal obesity and protection for T2D. A particular example of a peculiar metabolic discordance was revealed at the 2p21 locus containing *THADA* and *THADA-AS*, represented by variant rs6752964 (*Figure 4—figure supplement 2*). The associations have been replicated multiple times (*Mahajan et al., 2018*; *Zeggini et al., 2008*; *Grarup et al., 2008*), but the exact mechanisms underlying this association are unknown. *THADA* plays an evolutionarily conserved role in intracellular calcium signaling and consequently non-shivering thermogenesis. In *Drosophila melanogaster*, *thada* knockout flies developed obesity and hyperphagia without altered circulating glucose levels (*Moraru et al., 2017*). In mice, pancreatic *Thada* knockout resulted in protection from T2D through the preservation of β-cell mass and improvement of β-cell function (*Zhang et al., 2023*). Mendelian randomization studies in humans have likewise found consistent links between THADA and adiposity, but have not yet been able to link it to diabetic phenotypes such as insulin secretion (*Grarup et al., 2008*; *Simonis-Bik et al., 2010*). Our investigation revealed relationships between THADA and THADA-AS expression with diabetic and obesity-abdominal obesity phenotypes, as well as dynamic expression in adipogenesis (*Figure 3*). Regulatory interactions whereby *THADA-AS* expression interferes with *THADA* transcription could provide a basis by which variant rs6752964 might confer abdominal obesity, but protection from T2D (*Brantl, 2002*; *Faghihi and Wahlestedt, 2009*; *Wight and Werner, 2013*). Further, we also found colocalization of genetic regulation of *PEPD* in adipose tissue with the discordant association signal represented by variant rs3786897. Depletion of *PEPD* in preadipocytes has been shown to reduce adipogenic potential, decrease triglyceride accumulation, and phospho-Akt signaling, which is critical to insulin sensitivity (*Chen et al., 2020b*). Notably, a secondary signal represented by variant rs731839 was apparent in this locus but was not significant for WHRadjBMI. This signal has been associated with sex-specific effects on serum lipid levels in Han and Mulao populations (*Lin et al., 2014*). Further in vivo and in vitro work must be done to resolve this multi-tissue, multi-eGene locus.

Although our analysis incorporated genome, transcriptome, epigenome, and phenome data in multiple cohorts, and used the consensus of orthogonal methods to predict the mechanisms of action at discordant loci, follow-up is required to validate each prediction. Additionally, our genetic expression data used single-strand sequencing, and therefore parsing out the associated effects of sense and antisense transcripts is difficult. Finally, it is critical to discover to diversify ancestry and sex in genetic association studies to identify more genetic loci underlying MHO. Without experimental follow-up and extensive clinical studies, genotype should not be used as a diagnostic metric. CRISPR editing of alleles in relevant cell types to study *cis*-regulatory effects on genes and phenotypic effects on cells, and work in animal models is necessary to fully annotate these loci. In addition, it is important to identify the indirect and direct effects of discordant variants, as these endocrine tissues are major contributors to peptide and hormone secretion. Further experimental characterization is critical to placing these results in the proper context and providing the basis for personalized interventions for T2D. The predictions at these six signals provide specific hypotheses to be tested, and should they be validated experimentally provide knowledge of the precise mechanisms of uncoupling obesity from T2D risk.

## Methods
### GWAS-GWAS colocalization analysis

GWAS results for T2DadjBMI and WHRadjBMI were obtained from *Mahajan et al., 2018*, and *Pulit et al., 2019*. The set of single nucleotide polymorphisms (SNPs) within 500 kb of a genome-wide significant SNP in either GWAS was included in the colocalization test. Rare variants, defined as SNPs reported to have effect allele frequencies of less than 1% in either GWAS, were excluded. Proximal analysis windows (>250 kb) were merged, and the colocalization test was performed on these genetic loci with three methods: Coloc.abf (*Wallace, 2020*), Hyprcoloc (*Foley et al., 2021*), and visual inspection of LocusCompare plots (*Liu et al., 2019*).

The default parameters were used for Hyprcoloc. In Coloc.abf, the default parameters for p1 and p2 prior probabilities were used for the individual GWAS hypotheses. The parameter p12, the prior for single variant colocalization, was set to 5e-06 as prescribed by *Wallace, 2020*, to balance false negative and positive results. Loci were considered colocalized if the regional probability of colocalization was greater than 0.70. In Coloc.abf, this was the sum of the PPH3 and PPH4 statistics, and in Hyprcoloc this was the regional probability statistic. Loci that met colocalization criteria in either method were plotted using LocusCompare with the default European ancestry linkage disequilibrium (LD) data from 1000 Genomes (*Fairley et al., 2020*) and with genome build hg19. This resulted in 121 LocusCompare plots on which visual inspection was performed to verify colocalized genetic association signals. If genetic loci were considered colocalized by at least two of the three colocalization analysis methods, we considered these consensus colocalized loci. We termed this consensus analysis 'COLOC'.

## Discordant locus identification

We obtained the 99% credible set of SNPs from the results of Bayesian factor analysis implemented through Coloc.abf at each locus. We calculated the Z-scores for the association test of each genetic variant and the GWAS trait. If the Z-score associated with SNP had the opposite sign for association with WHRadjBMI and T2DadjBMI with respect to the same allele and the p-value for the association with both traits was less than 1e-05, we considered the variant discordant. We then identified in which loci the SNPs were located, and queried haploReg (*Ward and Kellis, 2016*) LD data with the haploR package in R (*Zhbannikov et al., 2017*) to separate signals in the same loci using LD clumping ($R^2 > 0.50$) on the discordant variants.

## Phenome-wide association study

We queried the GWAS meta-analysis associations of glycemic and anthropometric traits for each lead discordant variant in the Type 2 Diabetes Knowledge Portal (T2DKP) (*Costanzo et al., 2023*). We additionally obtained the summary statistics of abdominal fat MRI scans in the UK Biobank and queried these summary statistics for discordant variants (*Liu et al., 2021*).

## Multi-trait colocalization analysis

We obtained GWAS summary statistics for WC, hip circumference (HC), WHR, WHRadjBMI, T2D, and T2DadjBMI. We extracted summary statistics of variants within genetic loci containing a discordant association signal (*Mahajan et al., 2018*; *Pulit et al., 2019*) and performed multi-trait colocalization with Hyprcoloc (*Foley et al., 2021*). We considered an association signal colocalized for multiple traits if Hyprcoloc computed a posterior probability for both body fat distribution traits (WC, HC, WHR, and WHRadjBMI) and T2D or T2DadjBMI.

## Fine-mapping analysis

We performed variable selection in multiple regression as implemented in the R package SuSiE (*Wang et al., 2020*). This method implements the sum of single-effects models to fine-map the causal variant(s) in a locus. Using the T2DadjBMI and WHRadjBMI GWAS summary statistics and the 1000 Genomes LD data, we performed fine-mapping of loci containing a genetic variant associated with discordant effects on T2DadjBMI and WHRadjBMI. We used the default flag options in SuSiE and performed a sensitivity analysis of the results to a range of priors. We selected causal variants with a PPH4 greater than 0.70.

## GWAS-QTL colocalization analysis

We obtained eQTL data from the Genotype-Tissue Expression (GTEx) for 49 tissues (*Battle et al., 2017*), the Stockholm-Tartu Atherosclerosis Reverse Networks Engineering Task (STARNET) cohort for 6 tissues (*Franzén et al., 2016*), and the Metabolic Syndrome in Men (METSIM) for SAT (*Brotman et al., 2022*). We also obtained SAT sQTL results from the METSIM cohort. Data sources and further information are detailed in *Figure 2—source data 1*. We extracted the QTL data for each gene or transcript within 1 Mb of a discordant locus start or end site and independently colocalized with the T2DadjBMI and WHRadjBMI GWAS using Coloc.abf and SMR. When implementing Coloc.abf, we considered a signal to be colocalized if PPH4 was greater than 0.50 (a threshold used for GWAS-QTL

colocalization in admixed populations; *Gay et al., 2020*). We repeated the analysis in SMR and used an FDR threshold of 5% to control for false positives. We then performed a visual inspection of GWAS-QTL colocalization of plots generated by LocusCompare. If a GWAS-QTL colocalization met these criteria, the proximal gene was termed an eGene.

## fGWAS annotation enrichment analysis

We used the functional GWAS (fGWAS) (*Pickrell, 2014*) command-line tool to compute the enrichment of associations, in particular genomic and epigenomic regions. We first obtained the chromosome and base-pair position of each variant in the 99% credible set from each of the 79 colocalized loci. We mapped the SNPs to their placement in genomic regions using bed files. We used bed files from tissue-specific chromatin state data (adipose, liver, pancreatic islet, and skeletal muscle) and genome-level coding region annotations, and mapped SNPs to their presence in these regions. From these maps, we performed enrichment analysis with the complete model of all annotations with the -fine and -xv flags on fGWAS. We used the natural log of the Bayes factor of the colocalization test and computed the enrichment of SNPs for presence in coding regions to genetic and epigenetic annotations.

## TOA analysis

We conducted TOA score analysis using the credible set of SNPs from each of the 79 colocalized loci. TACTICAL computes the TOA score with the SNP-level Bayesian probabilities, the SNP annotation maps, and the annotation enrichment scores. We used the Coloc.abf PPH4 scores for the SNP-level Bayesian probability, the fGWAS annotation enrichment scores, and the SNP annotation maps to compute the TOA score at all colocalized loci. We separated independent association signals in the same loci (LD <0.5) with HaploReg (*Ward and Kellis, 2016*). With TACTICAL (*Torres et al., 2020*), we integrated the credible set of SNPs with the enrichment for genome-level and tissue-specific annotations. We used the default tissue classification thresholds of 0.20 to classify signals as belonging to a particular TOA and less than 0.10 difference to classify signals as sharing TOA assignments between multiple tissues.

## Gene expression and phenotype correlation analysis

For each eGene, we computed the biweight midcorrelation and its significance, as implemented by the Weighted Genetic Coexpression Network Analysis (WGCNA) package (*Langfelder and Horvath, 2008*), between gene expression with metabolic phenotypes measured in the METSIM cohort (*Laakso et al., 2017*). We controlled for false positives with a 5% FDR threshold as implemented by the q-values package in R (*John, 2002*).

## Adipogenesis gene expression dynamics analysis

We obtained Simpson-Golabi-Behmel syndrome preadipocyte adipogenesis time series gene expression data from GEO (accession number GSE76131) (*Nassiri et al., 2016*). We evaluated the dynamic expression of each adipose tissue eGene by fitting the gene expression over time to a linear model and applying the likelihood ratio test (LRT) to compare the time-dependent models to time-independent null models. We considered an eGene to be dynamically expressed in adipogenesis if the p-value of the LRT was less than 0.05.

## Additional information

### Funding

| Funder | Grant reference number | Author |
|---|---|---|
| National Heart, Lung, and Blood Institute | 2T32HL007284-46 | Yonathan Tamrat Aberra |
| National Institute of Diabetes and Digestive and Kidney Diseases | R01 DK118287 | Mete Civelek |

| Funder | Grant reference number | Author |
|---|---|---|

The funders had no role in study design, data collection and interpretation, or the decision to submit the work for publication.

## Author contributions
Yonathan Tamrat Aberra, Conceptualization, Data curation, Software, Formal analysis, Funding acquisition, Validation, Investigation, Visualization, Methodology, Writing - original draft, Project administration, Writing - review and editing; Lijiang Ma, Resources, Data curation, Software, Formal analysis, Investigation, Methodology, Writing - review and editing; Johan LM Björkegren, Resources, Data curation, Supervision, Methodology, Project administration, Writing - review and editing; Mete Civelek, Conceptualization, Resources, Data curation, Supervision, Funding acquisition, Validation, Methodology, Project administration, Writing - review and editing

## Author ORCIDs
Yonathan Tamrat Aberra ⓘ http://orcid.org/0000-0002-6055-2291
Mete Civelek ⓘ http://orcid.org/0000-0002-8141-0284

## Decision letter and Author response
Decision letter https://doi.org/10.7554/eLife.79834.sa1
Author response https://doi.org/10.7554/eLife.79834.sa2

# Additional files

## Supplementary files
• Supplementary file 1. Credible set of genetic variants in T2DadjBMI and WHRadjBMI colocalized loci.
• Supplementary file 2. Finemapping analysis of association signals in loci containing discordant association signals.
• Supplementary file 3. Multi-trait colocalization of body fat distribution component traits in discordant loci.
• Supplementary file 4. Variant effect predictions of discordant variants.
• Supplementary file 5. Tissue of action scores at colocalized loci.
• Supplementary file 6. Significant eQTL and sQTL colocalization results in coloc.
• Supplementary file 7. Significant eQTL and sQTL colocalization results in SMR.
• MDAR checklist

## Data availability
The current manuscript is a computational investigation using publicly available data, so no data have been generated for this manuscript. All publicly obtained data sets are included in Supplementary Table 1. All analysis and figure-generating code uploaded to the following Github repository: https://github.com/aberrations/predicting-functional-mechanisms-discordant-loci, (copy archived at swh:1:rev:e457dc5b8ecdc6abb119730bc64014876e9d852e).

The following previously published datasets were used:

| Author(s) | Year | Dataset title | Dataset URL | Database and Identifier |
|---|---|---|---|---|
| Pulit SL, Stoneman C, Morris AP | 2018 | Meta-analysis of Body Fat Distribution GWAS | https://zenodo.org/record/1251813 | Zenodo, 10.5281/zenodo.1251813 |

*Continued on next page*

*Continued*

| Author(s) | Year | Dataset title | Dataset URL | Database and Identifier |
|---|---|---|---|---|
| GTEx Consortium | 2017 | GTEx Analysis V8 eQTL | https://console.cloud.google.com/storage/browser/gtex-resources/GTEx_Analysis_v8_QTLs/GTEx_Analysis_v8_eQTL_all_associations?pageState=(%22StorageObjectListTable%22:(%22f%22:%22%255B%255D%22))&prefix=&forceOnObjectsSortingFiltering=false | Google Cloud Platform, 10.1038/nature25160 |
| Varshney A, Scott LJ, Welch RP | 2017 | Chromatin state predictions by tissue type | https://theparkerlab.med.umich.edu/data/papers/doi/10.1073/pnas.1621192114/chromatin_states/ | Parker Lab Chromatin States, pnas.1621192114 |
| Raulerson CK, Ko A, Kidd JC | 2019 | METSIM eQTL | https://mohlke.web.unc.edu/data/ | FTP, 10.1016/j.ajhg.2019.09.001 |
| Koplev S | 2022 | Stockholm-Tartu Atherosclerosis Reverse Networks Engineering Task | http://starnet.mssm.edu/ | STARNET, 10.1038/s44161-021-00009-1 |
| Nassiri I | 2016 | Network Activity Score Finder analysis of SGBS cells | https://www.ncbi.nlm.nih.gov/geo/query/acc.cgi?acc=GSE76131 | NCBI Gene Expression Omnibus, GSE76131 |

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
