## [Editor Report]

This study presents a valuable finding on five genome-wide association study loci displaying discordant effects on T2D and abdominal obesity. The evidence supporting the claims of the authors is solid, although further experiments are required to describe the mechanisms through which a genetic variant can confer increased abdominal obesity while offering protection from T2D risk. The work will be of interest to researchers working within the fields of variant-to-function analysis and endocrinology.

---

## [Decision Letter]

**Decision letter after peer review:**

Thank you for submitting your article "Predicting mechanisms of action at genetic loci associated with discordant effects on type 2 diabetes and abdominal fat accumulation" for consideration by *eLife*. Your article has been reviewed by 3 peer reviewers, including Tune H Pers as the Reviewing Editor and Reviewer #1, and the evaluation has been overseen by David James as the Senior Editor. The following individual involved in review of your submission has agreed to reveal their identity: Marcel den Hoed (Reviewer #2).

Essential revisions:

1) Please carefully address, through additional analyses, the comments related to whether the proposed effector genes are likely "healthy obesity genes", or whether there are at these loci either multiple causal genes per locus, or single causal genes acting on different traits via different tissues and mechanisms.

2) Please carefully address the comments on WHRadjBMI being a trait that is complicated to interpret due to its composite structure, ideally by including analyses focusing on the two different fat depots that the WHR measure is based on.

3) Please address, where appropriate through additional analyses, all remaining comments raised by the three referees.

*Reviewer #2 (Recommendations for the authors):*

I read your manuscript with great interest and think the findings are interesting. Nevertheless, I hope the points raised below can help make the manuscript even better, and help support some of the conclusions drawn.

1. The trait that is WHRadjBMI has puzzled me for a while. Ratios can be problematic per se if two traits have a non-zero intercept (I am not sure if this is the case for WC and HC?), and adjusting for a heritable trait (BMI) may introduce collider bias, as was elegantly shown by Day et al. (https://pubmed.ncbi.nlm.nih.gov/26849114/). As a result, an association with WHRadjBMI may not reflect the same biological phenomenon for each locus. Hence, in addition to associations with WHRadjBMI and T2DadjBMI, it would be informative to also show summary statistics for associations of these loci with waist circumference, hip circumference, BMI and T2D in Figure 1.

2. By running a co-localization analysis with sets of SNPs within 200kb of a genome-wide significant SNP in either GWAS, the number of truly co-localizing signals may be underestimated due to secondary association signals in either or both GWASs. It could be worthwhile to additionally run conditional analyses for each outcome to identify such loci and run the co-localization analysis for each association signal separately.

3. The authors state that associations were considered discordant if the z-score had an opposite direction for the same allele. Presumably, the association with both traits also needed to be significant?

4. In a best-case scenario, WHRadjBMI indeed reflects abdominal obesity, as intended. However, associations with WHRadjBMI then reflect the net association with adiposity across all individual abdominal fat depots. This does not tell us which lipid depot(s) drive(s) the association with WHRadjBMI. Could the authors explore associations with e.g., liver fat and pancreas fat as derived by MRI in UK biobank data for these six loci?

5. Associations don't provide insights into causal pathways or directions of effect. Hence, I suggest refraining from using causal language, e.g., in line 75: "protecting against T2D but increasing abdominal fat accumulation". Even if a causal variant acts on both outcomes through a single causal gene and pathway (both are debatable at this stage), then there are still four possible courses of action. Similarly, in the discussion (line 237) the authors state that: "increased expression of THADA-AS results in increased abdominal fat accumulation but decreased T2D risk". No evidence was provided that justifies this conclusion based on an association.

6. Even if there is only one causal gene, its effects on different outcomes may be mediated by multiple tissues and cell types (and thus, mechanisms). Since all six loci showing directionally discordant associations with WHRadjBMI and T2DadjBMI are located in non-coding regions, these loci likely have a regulatory role, e.g., as enhancers. Hence, the seemingly directionally discordant associations may well be mediated by more than one causal gene. I would argue that neither the "one gene, multiple pathways" nor the "multiple genes" scenarios reflect uncoupling.

7. Co-localization of credible variants with sQTLs and eQTLs does not mean that such genes are causal for the traits of interest, as the association between the expression of a gene X and the outcome of interest can be confounded by the expression of the truly causal gene Y, with the expression of both genes X and Y influenced by the enhancer that is flagged by the causal variant. To get one step closer to such inference, it would be informative if the authors could show chromatin-chromatin interaction data for credible variants in all six discordant loci, across all relevant cell types.

8. Gene prioritization is a fast-moving field, and a range of bioinformatics approaches – but no gold standard – are available to prioritize candidate genes. Results using proof-of-concept loci have not shown very convincing results for the predictive ability of eQTL associations to flag causal genes, even in the presence of co-localization. Hence, it would be informative to see which genes are prioritized in these loci by other available approaches – including but not limited to chromatin accessibility QTLs in relevant cell types – and if there is consensus across those approaches as to which gene(s) are likely causal.

9. Even if there are loci that uncouple genetic effects on abdominal adiposity and risk of type-2 diabetes, the causal beneficial effect of weight loss is so large in comparison that it would no doubt remain advisable for most if not all individuals to lose weight to reduce the risk of T2D and its complications.

10. Why was ATAC-sec data provided for only one of six discordant loci? In the discussion the authors state that tissues and mechanisms of action are predicted for six loci, but only one or two are described in the paper.

*Reviewer #3 (Recommendations for the authors):*

1. The rationale for the study is unclear. The authors introduce the study as an investigation into factors explaining individual differences in the relationship between obesity and cardiometabolic health, particularly the "metabolically healthy obesity" and "lean, insulin resistance" phenotypes. However, the analyses use outcome phenotypes that are adjusted for BMI (i.e. obesity): WHRadjBMI and T2DadjBMI. Thus, the results do not explain differences in obesity status and metabolic health, but rather tell about differences in body fat distribution and T2D risk, independent of obesity. Furthermore, a relatively low abdominal fat accumulation is one of the key characteristics of the "metabolically healthy obesity" phenotype associated with lower T2D risk, while the "lean, insulin resistance" phenotype associated with increased T2D risk, is characterized by relative increase in abdominal adiposity. Thus, the discordant association pattern of relatively higher abdominal adiposity but lower risk of T2D, investigated by the authors, does not exemplify either of these two phenotypes. Could the authors think of a biological or clinical example that could represent the discordant genetic association pattern they are aiming to characterize, in order to create a more plausible rationale for the present analyses?

2. Overall, it is difficult to imagine a plausible biological explanation for a relationship where higher WHRadjBMI would mediate a lower risk of T2DadjBMI. It would be really interesting is such biology exists, but at the moment the paper does not give much help to the reader in interpreting what biological mechanism could underlie such a relationship. An alternative explanation for the findings is the possibility that the genetic effects are driven by independent effects of the same genes on biological pathways in two separate tissues (e.g. adipose tissue and pancreas), which would then generate the seemingly discordant genetic association pattern between WHRadjBMI and T2DadjBMI.

3. A clear weakness in the study is the lack of any consideration to the subcomponents of the WHR phenotype: waist and hip circumference. As WHR is calculated as the ratio of waist circumference to hip circumference, WHR can increase if waist circumference increases or if hip circumference decreases, reflecting an increase in abdominal fat or a decrease in gluteofemoral fat, respectively. Waist and hip circumferences have very different implications for cardiometabolic health and T2D risk. Thus, is it critical to include an analysis on genetic effects on the two different fat depots, to understand the biological underpinnings of the findings. In addition, further GWAS results could be used to look into more detailed fat distribution phenotypes, such as the amount of visceral fat and subcutaneous fat in different body compartments.

4. The results show that two of the independent discordant loci (represented by rs3776705 and rs459193), associated with lower risk of T2DadjBMI, show genome-wide significant associations with higher 2-hour glucose levels, and one of the loci (rs459193) also shows a genome-wide significant association with higher fasting glucose levels, despite their protective effect on T2D risk. The findings appear counter-intuitive, considering that T2D is diagnosed based on elevated fasting and 2-hour glucose values. Particularly for these two loci, it would be critical to make sure that the directions of effects and the reported reference alleles are validated in a further set of data, to make sure that the discordant effects are not due to mismatched alleles.

5. The paper includes several overstated claims about the clinical impact of the findings and use of the findings for personalized risk stratification [e.g. lines 77-78, 259-263]. The authors should revise the manuscript to avoid misleading the readers, by overstating the implications of their findings.

---

## [Author Response]

Essential revisions:1) Please carefully address, through additional analyses, the comments related to whether the proposed effector genes are likely "healthy obesity genes", or whether there are at these loci either multiple causal genes per locus, or single causal genes acting on different traits via different tissues and mechanisms.

We appreciate the suggestions for additional analysis to determine the proposed effector disease genes. To increase confidence in effector genes, we performed summary-based randomization (SMR) using T2DadjBMI and WHRadjBMI with QTLs. This provided an orthogonal method for colocalization that infers the causality of QTLs for GWAS associations. To control for false positives associated with MR methods, we implemented a false-discovery rate threshold of 5e-02 and performed a visual inspection of the association signal as visualized by LocusCompare. Implementing a requirement that a single variant that is associated with both the eQTL and trait of interest yielded 20 significant eQTL colocalizations, which have now been reported in Supplementary File 7. Additionally, we updated the table labels in the supplementary files to differentiate between colocalization results achieved using Coloc.abf and using SMR (Supplementary Files 6 & 7). We have updated our interpretation in the results to reflect the potential causal genes as either single effector genes, or multiple causal genes per locus acting on different traits (lines 161-190, 227-252, and 254-264).

2) Please carefully address the comments on WHRadjBMI being a trait that is complicated to interpret due to its composite structure, ideally by including analyses focusing on the two different fat depots that the WHR measure is based on.

We appreciate the comments on the choice of WHRadjBMI as a trait. To increase the interpretability of our analysis, we performed a multi-trait colocalization analysis for each of the loci containing a discordant association signal in the SNP set using the summary statistics for Waist Circumference (WC), Hip Circumference (HC), the ratio of Waist-to-Hip circumference (WHR), Body-Mass Index (BMI), WHR adjusted for BMI (WHRadjBMI), Type 2 Diabetes (T2D), and T2D adjusted for BMI (T2DadjBMI). In this recursive multi-trait colocalization analysis, traits are pairwise analyzed, and then summary statistics from traits are successively added and removed until the highest likelihood is achieved. This is repeated until all the colocalization of genetic association signals for each trait is tested and the maximal number of high-confidence colocalized groups of traits at each locus is achieved. This analysis was able to replicate colocalization at all six discordant loci. Further, at three of the six discordant loci the WHRadjBMI association signal was colocalized with the WC association signal. The colocalization analysis with the additional traits is now summarized in Supplementary File 3. The text has been updated to reflect these results (lines 123-127) and the methods (lines 353-358). In addition, we performed a comprehensive phenome-wide association study (PheWAS) of the individual abdominal fat depots underlying WHRadjBMI as well as the molecular glycemic phenotypes underlying T2DadjBMI (Figure 1C). The text has been updated to reflect the results of this PheWAS (lines 108-144).

3) Please address, where appropriate through additional analyses, all remaining comments raised by the three referees.

We appreciate the comments raised by referees and feel they have significantly improved the quality of the manuscript. We aimed to respond to each comment raised by referees comprehensively and have included responses to all comments below.

Reviewer #2 (Recommendations for the authors):I read your manuscript with great interest and think the findings are interesting. Nevertheless, I hope the points raised below can help make the manuscript even better, and help support some of the conclusions drawn.1. The trait that is WHRadjBMI has puzzled me for a while. Ratios can be problematic per se if two traits have a non-zero intercept (I am not sure if this is the case for WC and HC?), and adjusting for a heritable trait (BMI) may introduce collider bias, as was elegantly shown by Day et al. (https://pubmed.ncbi.nlm.nih.gov/26849114/). As a result, an association with WHRadjBMI may not reflect the same biological phenomenon for each locus. Hence, in addition to associations with WHRadjBMI and T2DadjBMI, it would be informative to also show summary statistics for associations of these loci with waist circumference, hip circumference, BMI and T2D in Figure 1.

We appreciate the comments on the choice of WHRadjBMI as a trait. To increase the interpretability of our analysis, we performed a multi-trait colocalization analysis for each of the loci containing a discordant association signal in the SNP set using the summary statistics for Waist Circumference (WC), Hip Circumference (HC), the ratio of Waist-to-Hip circumference (WHR), Body-Mass Index (BMI), WHR adjusted for BMI (WHRadjBMI), Type 2 Diabetes (T2D), and T2D adjusted for BMI (T2DadjBMI). In this recursive multi-trait colocalization analysis, traits are pairwise analyzed, and then summary statistics from traits are successively added and removed until the highest likelihood is achieved. This is repeated until all the colocalization of genetic association signals for each trait is tested and the maximal number of high-confidence colocalized groups of traits at each locus is achieved. This analysis was able to replicate colocalization at all six discordant loci. Further, at three of the six discordant loci the WHRadjBMI association signal was colocalized with the WC association signal. The colocalization analysis with the additional traits is now summarized in Supplementary File 3. The text has been updated to reflect these results (lines 123-127) and the methods (lines 353-358). In addition, we performed a comprehensive phenome-wide association study (PheWAS) of the individual abdominal fat depots underlying WHRadjBMI as well as the molecular glycemic phenotypes underlying T2DadjBMI (Figure 1C). The text has been updated to reflect the results of this PheWAS (lines 108-144). Further, the point is excellently taken about the collider bias, and this suggestion has increased the quality of our manuscript.

2. By running a co-localization analysis with sets of SNPs within 200kb of a genome-wide significant SNP in either GWAS, the number of truly co-localizing signals may be underestimated due to secondary association signals in either or both GWASs. It could be worthwhile to additionally run conditional analyses for each outcome to identify such loci and run the co-localization analysis for each association signal separately.

We apologize for the confusion, and the Methods section has been updated to reflect this. SNP windows of 200kb were constructed and then merged with nearby windows to form the loci that underwent colocalization analysis. This was to include all variants in linkage disequilibrium with the lead signal at the locus, as well as to include secondary association signals. To determine if the initial SNP window of 200kb significantly altered results, we altered the initial SNP window to 500kb and did not discover new loci containing discordant association signals, or find that previously identified loci were no longer significant. We amended the text in the Methods (line 327) to reflect this change.

Additionally, we implemented the sum of single effects (SuSiE) model to relax the single causal variant assumption and parse the effects of potential multiple causal signals at a locus. In conjunction with colocalization analysis, this found a separate discordant association signal in the 5q21.2 locus that was even more strongly associated with genetic regulation than the primary signal identified in our earlier analysis (Figures 1, 2, and 4). We report our results in Supplementary File 3 and throughout the Results section. We again thank the reviewers for this particularly helpful suggestion that has improved the quality of our manuscript.

3. The authors state that associations were considered discordant if the z-score had an opposite direction for the same allele. Presumably, the association with both traits also needed to be significant?

This is an excellent point, and our methods section has been updated to clarify this (lines 332 – 341). We additionally added all of the p-values for the association to Supplementary File 1 to further clarify the significance of association within the 99% credible set of SNPs. The association with both traits needed to reach at least a nominal, locus-wide significance of 5e-04. The signals for both WHRadjBMI and T2DadjBMI at four of the five discordant loci reached a genome-wide significance level of 5e-08. At the locus represented by variant rs74567345, the association p-value for WHRadjBMI was 1.599e-05. At the locus represented by the variant rs3786897, the association p-value for T2DadjBMI was 1.6e-07. Because these loci passed visual inspection as well as having significant Bayesian factors (>0.70) we considered them for further analysis. Additionally, while genome-wide significance is traditionally used to prioritize variant candidates for further investigation, work done in the laboratory of Brian W. Parks (PMID: 32197071) demonstrates that even variants with only nominal, local significance in GWAS can also have functional relevance to GWAS traits. We additionally used SuSiE to parse the colocalization of multiple causal signals in loci and found the association signal represented by variant rs6860588 that did reach genome-wide significance for association with both traits, as described in the Methods section and the response to reviewers above.

4. In a best-case scenario, WHRadjBMI indeed reflects abdominal obesity, as intended. However, associations with WHRadjBMI then reflect the net association with adiposity across all individual abdominal fat depots. This does not tell us which lipid depot(s) drive(s) the association with WHRadjBMI. Could the authors explore associations with e.g., liver fat and pancreas fat as derived by MRI in UK biobank data for these six loci?

We thank the reviewers for an excellent suggestion here. We followed this line of inquiry and reported the results in Figure 1C in response to the concerns raised above. This suggestion improved our results. The text has been updated to reflect these changes (lines 349-352).

5. Associations don't provide insights into causal pathways or directions of effect. Hence, I suggest refraining from using causal language, e.g., in line 75: "protecting against T2D but increasing abdominal fat accumulation". Even if a causal variant acts on both outcomes through a single causal gene and pathway (both are debatable at this stage), then there are still four possible courses of action. Similarly, in the discussion (line 237) the authors state that: "increased expression of THADA-AS results in increased abdominal fat accumulation but decreased T2D risk". No evidence was provided that justifies this conclusion based on an association.

We thank the reviewers for their comments on this topic in particular. We have changed the language we used in those specific lines and throughout the manuscript to reflect greater caution with causal language. We have moved to language that proposes a model that reconciles discordant effects and emphasized in the Discussion the need for experimental validation and further studies to conclusively show that discordant loci eGene expression or splicing are causal (lines 312-324).

6. Even if there is only one causal gene, its effects on different outcomes may be mediated by multiple tissues and cell types (and thus, mechanisms). Since all six loci showing directionally discordant associations with WHRadjBMI and T2DadjBMI are located in non-coding regions, these loci likely have a regulatory role, e.g., as enhancers. Hence, the seemingly directionally discordant associations may well be mediated by more than one causal gene. I would argue that neither the "one gene, multiple pathways" nor the "multiple genes" scenarios reflect uncoupling.

We thank the reviewer for this comment, and we added it to the Discussion section (lines 360-380) to reflect this uncertainty of interplay between variants, genes, and tissues. We also removed causal language throughout the manuscript for metabolically healthy obesity. Because of the directionally discordant associations, we do still believe the variants confer protection from diabetes but increased abdominal obesity and propose models to reconcile this.

7. Co-localization of credible variants with sQTLs and eQTLs does not mean that such genes are causal for the traits of interest, as the association between the expression of a gene X and the outcome of interest can be confounded by the expression of the truly causal gene Y, with the expression of both genes X and Y influenced by the enhancer that is flagged by the causal variant. To get one step closer to such inference, it would be informative if the authors could show chromatin-chromatin interaction data for credible variants in all six discordant loci, across all relevant cell types.

We thank the reviewers for this suggestion– we thought following these lines of inquiry would significantly improve the quality of our manuscript. We obtained the chromatin-chromatin interaction data from Hi-C experiments performed on adipocytes, skeletal myocytes, and pancreatic islets. We were unable to obtain interaction data from Hi-C experiments performed on hepatocytes. We then searched for the presence of variants in our credible set in the “bait” and “prey” regions of chromatin and were unable to discover the presence of discordant variants in the credible set within regions containing significant chromatin-chromatin interactions. While this revealed significant associations in the loci of interest, none of the discordant variants were in “bait” or “prey” regions in any cell line or tissue of interest. In summary, we concluded that chromatin-chromatin interactions are not a significant driver at the QTL colocalizations. We report the studies we obtained chromatin-chromatin interaction data from in Figure 2 – source data 2 and the results of our analysis in lines 156-157.

8. Gene prioritization is a fast-moving field, and a range of bioinformatics approaches – but no gold standard – are available to prioritize candidate genes. Results using proof-of-concept loci have not shown very convincing results for the predictive ability of eQTL associations to flag causal genes, even in the presence of co-localization. Hence, it would be informative to see which genes are prioritized in these loci by other available approaches – including but not limited to chromatin accessibility QTLs in relevant cell types – and if there is consensus across those approaches as to which gene(s) are likely causal.

We appreciate this comment from the reviewer. We used Summary-Based Mendelian randomization to prioritize eGenes with a direct association between expression levels and traits of interest. This orthogonal method to Coloc.abf provided us with more significant associations and replicated a limited set from Coloc.abf (Supplementary Files 6 and 7). We considered looking into chromatin accessibility QTLs (caQTLs) in relevant cell types as well, but we were only able to obtain liver caQTL data. Because we did not propose mechanisms of action in the liver for our discordant loci, we decided this would not represent a further prioritization or more association data to support or discredit our hypotheses and did not move forward with this. Nonetheless, we thank the reviewer for this suggestion as this did prompt us to look more critically at publicly available data to support and prioritize our candidate genes. We decided instead to look at time series preadipocyte differentiation data to see if our candidate eGenes played a role in adipogenesis (Figure 3B). This revealed significant associations with dynamic expression across adipogenesis. Because adipogenesis is known to be a significant molecular regulator of adipose tissue distribution, this helped us to prioritize candidate genes and form hypotheses at the 2p21 and 5q21.2 loci.

9. Even if there are loci that uncouple genetic effects on abdominal adiposity and risk of type-2 diabetes, the causal beneficial effect of weight loss is so large in comparison that it would no doubt remain advisable for most if not all individuals to lose weight to reduce the risk of T2D and its complications.

We agree with the reviewer, and thank them for this comment. We have added clarifying language in the introduction (lines 53-106) and Discussion section (lines 277-291) to further reflect the value of our investigation in the proper context of basic science research into mechanisms and not clinical advice.

10. Why was ATAC-sec data provided for only one of six discordant loci? In the discussion the authors state that tissues and mechanisms of action are predicted for six loci, but only one or two are described in the paper.

We thank the reviewer for this suggestion, and have removed the ATAC-Seq results from our manuscript.

Reviewer #3 (Recommendations for the authors):1. The rationale for the study is unclear. The authors introduce the study as an investigation into factors explaining individual differences in the relationship between obesity and cardiometabolic health, particularly the "metabolically healthy obesity" and "lean, insulin resistance" phenotypes. However, the analyses use outcome phenotypes that are adjusted for BMI (i.e. obesity): WHRadjBMI and T2DadjBMI. Thus, the results do not explain differences in obesity status and metabolic health, but rather tell about differences in body fat distribution and T2D risk, independent of obesity. Furthermore, a relatively low abdominal fat accumulation is one of the key characteristics of the "metabolically healthy obesity" phenotype associated with lower T2D risk, while the "lean, insulin resistance" phenotype associated with increased T2D risk, is characterized by relative increase in abdominal adiposity. Thus, the discordant association pattern of relatively higher abdominal adiposity but lower risk of T2D, investigated by the authors, does not exemplify either of these two phenotypes. Could the authors think of a biological or clinical example that could represent the discordant genetic association pattern they are aiming to characterize, in order to create a more plausible rationale for the present analyses?

We thank the reviewer for their suggestion. We aimed to incorporate this into our Discussion section in lines 254-264 and 277-291 and reflect caution with strong claims. We additionally have amended our introduction to put our findings in the context of the field, describing progress made to describe MHO loci that act on overall obesity, but not on loci associated with abdominal obesity but with protection from type 2 diabetes (lines 76-83). Finally, we found literature evidence supporting our hypothesis at the 2p21 locus containing the gene encoding THADA, which we expand upon in the discussion (lines 292-305).

2. Overall, it is difficult to imagine a plausible biological explanation for a relationship where higher WHRadjBMI would mediate a lower risk of T2DadjBMI. It would be really interesting is such biology exists, but at the moment the paper does not give much help to the reader in interpreting what biological mechanism could underlie such a relationship. An alternative explanation for the findings is the possibility that the genetic effects are driven by independent effects of the same genes on biological pathways in two separate tissues (e.g. adipose tissue and pancreas), which would then generate the seemingly discordant genetic association pattern between WHRadjBMI and T2DadjBMI.

We agree with the reviewer, and have attempted to interpret our results and propose a biological explanation for two of our five loci where we were able to reconcile our results. The text has been amended to include this, and our Figure 4 and Figure 4 – supplementary figure 2 are focused on this goal. The text has been amended in lines 297-321 and 361-381. We hope this addresses the concerns about biological plausibility. We additionally include literature evidence supporting our hypothesis at the 2p21 locus containing the gene encoding THADA, which we expand upon in the discussion (lines 292-305).

3. A clear weakness in the study is the lack of any consideration to the subcomponents of the WHR phenotype: waist and hip circumference. As WHR is calculated as the ratio of waist circumference to hip circumference, WHR can increase if waist circumference increases or if hip circumference decreases, reflecting an increase in abdominal fat or a decrease in gluteofemoral fat, respectively. Waist and hip circumferences have very different implications for cardiometabolic health and T2D risk. Thus, is it critical to include an analysis on genetic effects on the two different fat depots, to understand the biological underpinnings of the findings. In addition, further GWAS results could be used to look into more detailed fat distribution phenotypes, such as the amount of visceral fat and subcutaneous fat in different body compartments.

We appreciate the comments on the choice of WHRadjBMI as a trait. To increase the interpretability of our analysis, we performed a multi-trait colocalization analysis for each of the loci containing a discordant association signal in the SNP set using the summary statistics for Waist Circumference (WC), Hip Circumference (HC), the ratio of Waist-to-Hip circumference (WHR), Body-Mass Index (BMI), WHR adjusted for BMI (WHRadjBMI), Type 2 Diabetes (T2D), and T2D adjusted for BMI (T2DadjBMI). In this recursive multi-trait colocalization analysis, traits are pairwise analyzed, and then summary statistics from traits are successively added and removed until the highest likelihood is achieved. This is repeated until all the colocalization of genetic association signals for each trait is tested and the maximal number of high-confidence colocalized groups of traits at each locus is achieved. This analysis was able to replicate colocalization at all six discordant loci. Further, at three of the six discordant loci the WHRadjBMI association signal was colocalized with the WC association signal. The colocalization analysis with the additional traits is now summarized in Supplementary File 3. The text has been updated to reflect these results (lines 123-127) and the methods (lines 353-358). In addition, we performed a comprehensive phenome-wide association study (PheWAS) of the individual abdominal fat depots underlying WHRadjBMI as well as the molecular glycemic phenotypes underlying T2DadjBMI (Figure 1C). The text has been updated to reflect the results of this PheWAS (lines 108-144). Further, the point is excellently taken about the collider bias, and this suggestion has increased the quality of our manuscript. We appreciate the thoughtful suggestions.

4. The results show that two of the independent discordant loci (represented by rs3776705 and rs459193), associated with lower risk of T2DadjBMI, show genome-wide significant associations with higher 2-hour glucose levels, and one of the loci (rs459193) also shows a genome-wide significant association with higher fasting glucose levels, despite their protective effect on T2D risk. The findings appear counter-intuitive, considering that T2D is diagnosed based on elevated fasting and 2-hour glucose values. Particularly for these two loci, it would be critical to make sure that the directions of effects and the reported reference alleles are validated in a further set of data, to make sure that the discordant effects are not due to mismatched alleles.

We thank the reviewer for this suggestion. We found an error in the code that did flip the effect for fasting glucose for this locus and corrected it by pulling our PheWAS from the Type 2 Diabetes Knowledge Portal (T2DKP) directly instead of from individual manuscript summary statistics files. The effects now make sense in the context of protection from type 2 diabetes. We thank the reviewers for this suggestion and report our new results in Figure 1C.

5. The paper includes several overstated claims about the clinical impact of the findings and use of the findings for personalized risk stratification [e.g. lines 77-78, 259-263]. The authors should revise the manuscript to avoid misleading the readers, by overstating the implications of their findings.

We thank the reviewer for this suggestion, and have changed the specific language in lines 77-78 and 259-263. We have additionally revised the manuscript throughout to use greater caution with the implications of our findings.